# Satisfaction with Detrusor OnabotulinumtoxinA Injections and Conversion to Other Bladder Management in Patients with Chronic Spinal Cord Injury

**DOI:** 10.3390/toxins14010035

**Published:** 2022-01-03

**Authors:** Sheng-Fu Chen, Yuan-Hong Jiang, Jia-Fong Jhang, Hann-Chorng Kuo

**Affiliations:** Department of Urology, Hualien Tzu Chi Hospital, Buddhist Tzu Chi Medical Foundation, Tzu Chi University, Hualien 97004, Taiwan; madaux@yahoo.com.tw (S.-F.C.); redeemer1019@yahoo.com.tw (Y.-H.J.); alur1984@hotmail.com (J.-F.J.)

**Keywords:** neurogenic bladder, urinary incontinence, botulinum toxin, urodynamic study

## Abstract

This study investigated the satisfaction with continued detrusor Botox injections for urinary incontinence and conversion to other surgical procedures and bladder management procedures for neurogenic detrusor overactivity (NDO) in patients with chronic spinal cord injury (SCI). A total of 223 patients with chronic SCI underwent detrusor Botox 200U for urodynamically confirmed NDO and urinary incontinence. After initial detrusor Botox injections, patients opted to either continue detrusor Botox injections every six to nine months and on clean intermittent catheterization (CIC), switch to other bladder management procedures, or receive surgical procedures to improve their urinary incontinence, correct emergent complications, or have better voiding conditions without CIC. Urinary incontinence improvement rates and satisfaction with bladder management were assessed and compared between different subgroups, urodynamic parameters, and bladder management procedures. Finally, a total of 154 male and 69 female patients were included, among whom 56 (25.1%), 81 (36.3%), 51 (22.9%), and 35 (15.7%) showed a marked, moderate, mild, and no reduction in urinary incontinence, respectively. However, only 48.4% of the patients continued detrusor Botox injections over the mean follow-up period of seven years. Patients with cervical or thoracic SCI had fair incontinence improvement rates. The presence of high detrusor pressure and higher-grade bladder outlet resistance also predicted a decrease in incontinence. Although more than 50% of the patients switched to other bladder management procedures or received surgical treatment, 69.1% expressed satisfaction with their current status. This large cohort of patients with chronic SCI who received initial detrusor Botox injections revealed that only 48.4% continued with Botox injections. Those who received surgical procedures due to urological complications or demanded change in bladder management could achieve high satisfaction rates.

## 1. Introduction

Neurogenic detrusor overactivity (NDO) frequently occurs in patients with chronic spinal cord injury (SCI) due to interruption of spinal tract, lack of central nervous control and regulation, and emergence of C-fiber as the predominant afferent nerves [1]. Urinary incontinence, the most bothersome symptom of NDO, is usually difficult to completely eradicate after oral overactive bladder medication because current oral medication for NDO cannot completely block the detrusor’s contractility [2]. In patients who are still bothered by urinary incontinence, detrusor injection of onabotulinumtoxinA (Botox) can provide a chance to regain urinary continence [3].

Detrusor injection of Botox has an effect on preganglionic parasympathetic and adrenergic nerves and sensory fibres, blocking acetylcholine release and resulting in paralysis of motor neurons of the urinary bladder [3,4]. The human bladder muscarinic receptors M2 and M3 and purinergic receptors P2X3 and P2X3 are reduced after detrusor Botox injections [5]. Through inhibition of both sensory and motor nerves in the micturition reflex, the NDO in SCI bladders can be inhibited. Botox also alters the release of adenosine triphosphate, neurotrophins, and nitric oxide in the urothelium, which can also reduce sensory urgency both in neurogenic and non-neurogenic bladder dysfunction [3,6,7,8].

Several clinical trials and long-term follow-ups have proven detrusor botulinum toxin A (Botox) injection to be effective in treating urinary incontinence among patients with chronic SCI [9,10,11,12]. The incontinence grade and quality of life (QoL) can be significantly improved after detrusor Botox injections [9,10,11,12,13]. The therapeutic effectiveness of detrusor Botox injections has mainly been gauged by the reduction in incontinence episodes, increase in bladder capacity, and decrease in detrusor pressure [14,15]. Repeat Botox injections usually have similar treatment outcomes as the first one, with 200 U of onabotulinumtoxinA having been found to have a similar therapeutic efficacy as 300 U [14,16]. The therapeutic effects of Botox include not only amelioration of NDO and urinary incontinence but also a decreased grade of autonomic dysreflexia (AD) and urinary tract infection (UTI) [17,18]. 

However, patient adherence to detrusor Botox injection has been underwhelming, with over half of the patients with NDO discontinuing treatment within the first 10 years [19]. Patients might be troubled by the need for clean intermittent self-catheterization (CISC) or UTI after detrusor Botox injections [20]. Moreover, the necessity for repeat injections might prohibit approximately 40% of patients with SCI from continuing with this minimally invasive treatment, prompting them to switch to a more invasive but permanent surgical procedure for future bladder management [21]. 

This study therefore investigated the satisfaction with continuing detrusor Botox injections for urinary incontinence and conversion to other surgical procedures and bladder management procedures after the first detrusor Botox injection for NDO in patients with chronic SCI.

## 2. Results

Table 1 summarizes the patients’ demographic characteristics and improvement in urinary incontinence after initial Botox injection. Among the included patients, 154 and 69 were male and female, respectively, with the lesion site located at the cervical cord (*n* = 89), thoracic cord (*n* = 115), and lumbar and sacral cord (*n* = 19). Overall, 56 (25.1%), 81 (36.3%), 51 (22.9%), and 35 (15.7%) of the patients exhibited marked, moderate, mild, and no improvement in urinary incontinence, respectively. Patients with cervical and thoracic SCI had better urinary incontinence improvement rates. Those without AD and who were incapable of hand function also had better urinary incontinence improvement rates after detrusor Botox injections. 

The baseline urodynamic parameters are detailed in Table 2. Patients with moderate and marked improvement in urinary incontinence were younger, although the duration of SCI had no impact on improvement rates. The grade of urinary incontinence improvement increased as Pdet increased, FS decreased, US decreased, bladder compliance decreased, and BCI and BOOI increased. 

After classifying VUDS findings according to NDO with and without DSD or BND and the presence of DU and ISD, our results interestingly showed that patients having NDO without DSD (23.1% with moderate and marked improvement) and those having DU with low compliance (36.8%) or DU with ISD (0%) had less favorable treatment outcomes after detrusor Botox injection. Patients with high voiding pressure (>40 cmH_2_O) had better urinary incontinence improvement rates compared to those with low voiding pressure (*p* < 0.0001) (Table 3). When we analyzed the baseline urodynamic characteristics, more patients who discontinue Botox treatment had NDO without DSD or with DU, compared with those who continued Botox treatment (*p* = 0.039). These urodynamic results might result in a less favorable treatment outcome and influence decision making in future bladder management.

After the initial detrusor Botox injection, only 108 patients (48.4%) continued detrusor Botox injections for urinary control. A total of 88 patients (39.5%) did not continue with detrusor Botox injections and resumed spontaneous voiding with their previous bladder management. The causes for the discontinuation of Botox injections included UTI (41, 46.6%), burden of CIC/CISC (15, 17.0%), ease in voiding spontaneously (11, 12.5%), and dislike of injections under anesthesia (21, 23.9%). During the mean follow-up period of 7 years (5–15 years), 29 patients (13.0%) switched to AE, 24 (10.8%) received urethral Botox injection for difficulty in urination, 14 (6.3%) reverted back to an indwelling Foley catheter, 11 (4.9%) opted for CISC alone, and 8 (3.6%) changed to suprapubic cystostomy diversion. Moreover, 16 patients preferred voiding spontaneously and underwent TUI-BN (*n* = 9), TUI-P or TUR-P (*n* = 6), or external sphincterotomy (*n* = 1). One patient switched to an ileal conduit due to exacerbated AD during the follow-up period. The 88 patients who did not receive any further surgical procedure after the first detrusor Botox injection retained their ability to self-void through insensibility, reflex, percussion, or straining to void, among whom 53.4% also had significant improvement in incontinence after antimuscarinic medications (Table 4). Overall, 83 (37.2%) and 57 (25.6%) patients reported moderate and marked improvement in urinary incontinence severity.

After the initial detrusor Botox injection and following surgical procedure or bladder management, 154 (69.1%) patients reported satisfaction with their current treatment outcome, 40 (17.9%) reported acceptable outcomes but wished to change their bladder management if possible, and 22 (9.9%) were not satisfied with the treatment outcomes but did not want to change bladder management. Among the treatment modalities, the TUI-P/TUR-P (100%), AE (82.8%), detrusor Botox injection (75.9%), and ileal conduit and external sphincterotomy (all 100%) had the best satisfaction rates, followed by urethral Botox (62.5%), indwelling catheter (64.3%), CIC alone (63.6%), and suburethral sling (60%), with the remaining procedures or bladder management having a satisfaction rate of <60%. (Table 5)

## 3. Discussion

The current study revealed that 61.4% of patients with chronic SCI who had NDO exhibited a significant decrease in urinary incontinence (moderate and marked improvement) after detrusor Botox injections. However, only 48.4% of the patients continued with detrusor Botox injections during the mean follow-up period of 10 years. Patients with cervical or thoracic SCI displayed fair improvements in incontinence. The presence of high Pdet and higher-grade bladder outlet resistance also predicted a favorable decrease in incontinence. Although over 50% of the patients switched to other bladder management procedures or underwent surgery, 69.1% of the patients reported satisfaction with their current status.

Previous clinical trials have comprehensively documented the therapeutic efficacy of detrusor Botox in patients with SCI who had NDO [9,10,11,12]. Evidence has shown that patients with SCI who exhibit NDO and urinary incontinence usually experience increased bladder capacity and decreased incontinence amount and episodes after detrusor Botox injections, regardless of the level of SCI [22] and Botox injection dose (200 or 300 U) [23]. A 10-year follow-up of a large cohort of patients with SCI receiving detrusor Botox injections found that approximately 50% were still receiving injections for their NDO [24]. The results of this study confirmed that the persistence rate of detrusor Botox treatment for NDO in chronic SCI patients was consistent in different studies with long-term follow-up [19,20]. However, urinary retention was inevitable after detrusor Botox injections, necessitating CIC or CISC for periodic bladder emptying, which largely decreased the patients’ intention to continue with Botox treatment [25]. Patients who are able to void spontaneously via percussion or reflex might not appreciate this change in bladder management, given that CIC/CISC is quite a burden, especially in patients with normal and partial hand function [20]. Interestingly, we noted that patients with cervical SCI and those incapable of hand function showed a significant improvement in incontinence severity, mostly due to the availability of caregivers to perform CIC. Therefore, such patients reported a greater decrease in incontinence compared to the other subgroups. 

Previous studies had reported that a high baseline Pdet.max can usually predict poor urodynamic outcomes, whereas low bladder compliance was a predictor of failure after detrusor Botox injection [26,27]. Similarly, the current study found patients with a high Pdet and higher BCI and BOOI displayed marked improvement in urinary incontinence after detrusor Botox injection. This result indicates that a higher voiding pressure caused by greater bladder outlet resistance due to DSD or BND can effectively prevent urine leakage during detrusor overactivity contractions. However, caution should be taken, given that this high pressure might also exacerbate upper urinary tract function provided that CIC/CISC was not appropriately performed.

Considering that only 61.4% of the patients with SCI had significant improvement in urinary incontinence, such patients might opt to switch to other procedures for better urinary control. Patients might be recommended to undergo AE after repeat Botox injections due to persistent incontinence, minimal increase in bladder capacity, or limited reduction in intravesical pressure or compliance [28]. In fact, one study employing long-term follow-up showed that the QoL after AE was better than that of repeat Botox injections for chronic SCI patients [29]. Patients with remarkable BND or DSD might switch to TUI-BN, TUI-P or TUR-P, urethral Botox injection, or external sphincterotomy due to difficulty with CISC or to self-voiding by percussion or straining to void. Although 48.4% of the patients with SCI who continued with detrusor Botox injections exhibited a significant decrease in incontinence, those who received surgical procedures or switched to other modes of bladder management also showed good improvement in incontinence and overall satisfaction. Thus, early surgical intervention or other bladder management procedures should be considered in those who do not respond well to detrusor Botox injections.

Interestingly, although significant improvement in urinary incontinence was achieved in 64.1% of the patients with SCI, some patients complained that the need for CISC had become a burden to their daily life [20]. Patients with SCI who could void through reflex, percussion, or triggering might not appreciate this change in incontinence, especially those who had inadequate social support or difficulty finding a lavatory to perform CISC during their daily life or working hours. Some patients still had urinary incontinence, given that they could not perform CISC in time and had an over-distended bladder. Some patients with SCI might experience UTI for the first time after initial detrusor Botox injection. Therefore, 39.5% of the patients opted not to continue with detrusor Botox injections and resumed their previous bladder management without the need for CIC/CISC. This result is similar to that presented in other studies on long-term adherence to detrusor Botox injections in patients with SCI [19,24]. Detrusor Botox injections provided patients with SCI who had cervical SCI, quadriplegia, urinary incontinence, and incomplete bladder emptying with the opportunity to be rid of their urinary incontinence, which would improve their QoL and prompt them to continue with Botox injections and CIC by a caregiver. This could explain why patients with SCI who were incapable of hand function had higher urinary incontinence improvement rates. 

In actual clinical practice, patients with chronic SCI might develop several urological complications, such as recurrent UTI, exacerbated AD, hydronephrosis due to low bladder compliance, and severe dysuria and urinary incontinence [30]. Improvement in urinary incontinence usually did not strongly correlate with expectations of improvement in urination QoL [31]. Therefore, bladder management after surgical procedure would be necessary to satisfy their need for both bladder storage and emptying functions. The results were that satisfaction in patients following TURP/TUIP or external sphincterotomy was higher than satisfaction with the treatment of incontinence, suggesting that the bladder emptying phase was more important for their quality of life than the storage phase. Before suggesting detrusor Botox injection to patients, patients’ needs as related to lower urinary tract dysfunctions should be clearly identified to avoid selection bias.

AE with or without ureteral reimplantation has been frequently performed to treat hydronephrosis, vesicoureteral reflux, and recurrent acute pyelonephritis due to high intravesical pressure and low bladder compliance [28]. To regain self-voiding, patients with SCI might opt for TUI-P, TUR-P, external sphincterotomy, or urethral sphincter Botox injection to decrease bladder outlet resistance and resume spontaneous voiding via reflex, percussion, or abdominal straining [32]. Although detrusor Botox injection could reduce the severity of AD in patients with SCI, an ileal conduit diversion might be a better option over repeat Botox injections for those with severe AD and incontinence [33]. A sub-urethral sling in female patients with SCI could increase urethral resistance and decrease the severity of urinary incontinence. These surgical procedures can not only improve urinary incontinence but also increase patients’ overall satisfaction with bladder management [34].

Appropriate patient selection is the key for successful outcomes of detrusor Botox injection for NDO in chronic SCI patients. In this study, 40 (18%) patients opted for surgical procedures aiming at bladder emptying and not urine storage, and 88 (39.5%) finally decided to resume spontaneous voiding, suggesting that the treatment strategy, in the initial decision to select detrusor Botox injection, was wrong. Patients with SCI might have urinary incontinence with predominant emptying symptoms rather than storage symptoms. These patients who were chosen for detrusor Botox injection were unhappy with CISC, although they agreed to the required pretreatment counselling after Botox treatment. Because they could not accept CISC to empty their bladder, they were therefore highly likely to discontinue Botox treatment and switch back to their original bladder management or receive other surgical treatments. This result further implies the importance of patient education and counselling before Botox treatment.

After detrusor Botox injections and other surgical procedures, around 70% of the patients with SCI were satisfied with their current bladder management. Although approximately 10% of the patients were not satisfied, they usually accepted their voiding condition as a certain outcome of SCI, and therefore opted to continue with their current management. While patients managed with cystostomy or indwelling Foley catheter might consider changing their bladder management, they usually hesitated to receive more invasive surgical procedures. Indeed, a long-term survey unsurprisingly showed that more and more patients with SCI opted for indwelling catheter or cystostomy over CISC after SCI for more than five years [35]. In fact, adequate education regarding proper bladder management may take precedence over achieving urinary continence. Regular follow-up for renal and bladder function, adequate hydration, and sufficient medication to decrease intravesical pressure or urethral resistance might be necessary in patients with chronic SCI, especially those who opted for spontaneous voiding or CIC/CISC.

## 4. Conclusions

This long-term follow-up of a large cohort of patients with chronic SCI who received initial detrusor Botox injection revealed that only 48.4% of the patients continued with detrusor Botox injections. The results of this study have been biased by the inappropriate patient selection. In total, 39.5% of the patients did not continue due to adverse events, burden of CIC/CISC, dislike of intravesical injection, and inappropriate patient selection for detrusor Botox injection. Patients who received surgical procedures due to urological complications after detrusor Botox injections could achieve high satisfaction rates at long-term follow-up. Regular follow-up of renal function and voiding condition is imperative to avoid complications and improve QoL in patients with chronic SCI.

## 5. Materials and Methods

A total of 223 patients with chronic SCI received 200 U of detrusor Botox for urodynamically confirmed NDO and urinary incontinence. The patients had been treated and followed up in the urology department of the hospital from 2000 to 2020. All patients had >14 urinary incontinence episodes per week and failed to reduce incontinence episodes, despite receiving antimuscarinics. Before Botox injections, video urodynamic study (VUDS) was performed to assess the bladder condition during the storage phase and bladder outlet condition during the voiding phase. The following urodynamic parameters were determined: first sensation of filling (FSF), full sensation (FS), urge sensation (US), bladder compliance, cystometric bladder capacity (CBC), voiding detrusor pressure (Pdet), maximum flow rate (Qmax), post-void residual volume (PVR), bladder neck dysfunction (BND), detrusor sphincter dyssynergia (DSD), presence of detrusor underactivity (DU) and intrinsic sphincter deficiency (ISD), voiding efficiency (VE, voided volume/CBC), bladder contractility index (BCI, Pdet + 5 × Qmax), and bladder outlet obstruction index (BOOI, Pdet − 2 × Qmax). All VUDS procedures and parameters were conducted in accordance with the recommendations of the International Incontinence Society [36]. 

The Botox injection technique and follow-up protocol have been reported previously [37]. After receiving Botox injections, all patients needed CISC or clean intermittent catheterization (CIC) performed by their caregiver. They were regularly followed up every three to six months at the urological clinic to evaluate their voiding conditions. When the episodes and severity of urinary incontinence returned to baseline condition, patients were recommended to receive repeat detrusor Botox injection with the same (200 U) or increased (300 U) dose, at least six months apart from the prior detrusor Botox injection.

After the initial detrusor Botox injections, some patients opted to continue detrusor Botox injections every six to nine months along with CIC/CISC, switch to other bladder management procedures (indwelling urethral Foley catheter, cystostomy, or CIC/CISC without Botox injection), receive surgical procedures to improve their urinary incontinence, correct emergent complications (through augmentation enterocystoplasty (AE), ileal conduit, or sub-urethral sling), or have better voiding conditions without CIC/CISC (through urethral Botox injection, transurethral incision of bladder neck (TUI-BN), transurethral incision of prostate (TUI-P), transurethra resection of prostate (TUR-P), or external sphincterotomy). During the follow-up period, some patients might have received more than one surgical procedure to correct urological complications and improve their bladder and voiding conditions. Patients were continuously followed up at the urological clinic for evaluation and treatment. The treatment flow chart is presented in Figure 1.

Patients were assessed for decrease in urinary incontinence by their subjective perception of the symptom severity after detrusor Botox injections, surgical procedures, and other bladder management procedures during the preceding year. The final treatment outcome was assessed by chart review or telephone interview by the end of 2020. The improvement in urinary incontinence was graded as follows: (1) not improved (i.e., no significant change in the episodes and quantity of urinary incontinence); (2) mildly improved (i.e., reduction in urinary incontinence episode and diaper use by approximately 1/2 that of the previous amount); (3) moderately improved (i.e., reduction in urinary incontinence episode and diaper use by approximately 1/4 that of the previous amount); and (4) markedly improved (i.e., nearly dry or only one diaper used per day).

Moreover, patients were also assessed according to overall satisfaction with their treatment outcomes in both storage and emptying conditions after detrusor Botox injections, surgical procedures, and other bladder management procedures. Overall satisfaction was graded as follows: (1) very satisfied (i.e., marked improvement in QoL and satisfaction with urinary control and bladder emptying); (2) satisfied but wished for change (i.e., moderate improvement in QoL albeit with slightly bothersome urinary control or bladder emptying); (3) satisfied but no change demanded (i.e., no change in treatment wanted despite moderate improvement in QoL with considerably bothersome urinary control or bladder emptying); and (4) not satisfied (i.e., no improvement in QoL and bladder management after treatment).

The patients’ demographics, urodynamic characteristics, and surgical procedures and bladder management were recorded and compared according to urinary incontinence improvement grade and overall satisfaction with the surgical procedures and bladder management. Patients who were lost to follow-up and the data unavailable to assess were not included in this study. This study had been approved by the Research Ethics Committee of Hualien Tzu Chi Hospital (IRB: 110-033-B). Given the retrospective nature of this study, the requirement for informed consent was waived by the Research Ethics Committee of Hualien Tzu Chi Hospital. All methods used in this study were conducted in accordance with relevant guidelines and regulations.

Continuous variables were expressed as the mean and standard deviation, whereas categorical data were presented as numbers and percentages. The chi-square test for categorical variables and Wilcoxon rank-sum test for continuous variables were used to determine differences between groups. All statistical assessments were two-sided and considered significant at *p* < 0.05. All calculations were performed using Statistical Product and Service Solution for Windows (version 16.0, Chicago, IL, USA). 

## Figures and Tables

**Figure 1 toxins-14-00035-f001:**
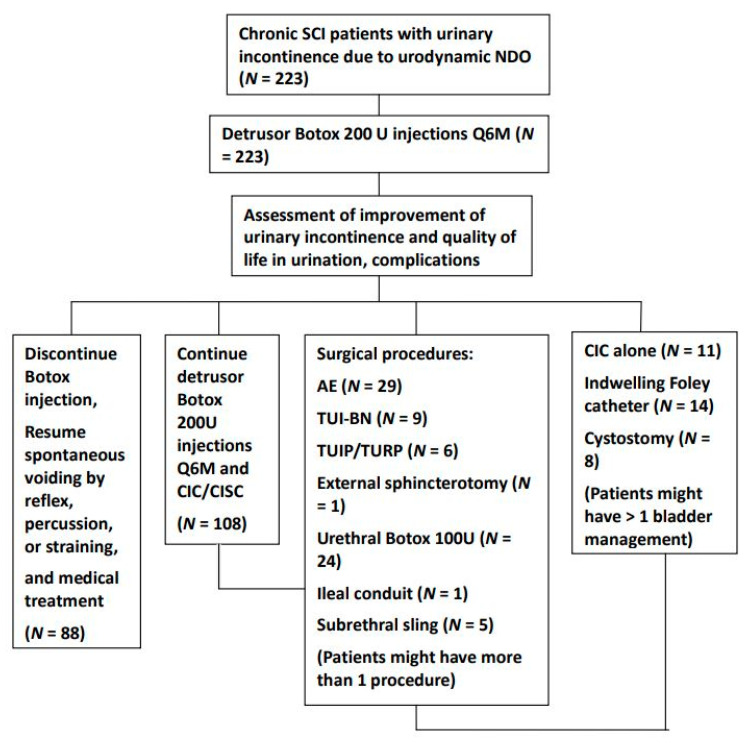
Treatment flow chart detailing the bladder management and surgical procedures in patients with chronic spinal cord injury who had neurogenic detrusor overactivity after detrusor botulinum toxin A injection. During the follow-up period, patients might receive several different surgical procedures and bladder management. Assessments were performed the year after the procedure. (Abbreviations: SCI: spinal cord injury, CIC: clean intermittent catheterization, CISC: clean intermittent self-catheterization, AE: augmentation enterocystoplasty, TUI-BN: transurethral incision of bladder neck, TUI-P: transurethral incision of the prostate, TUR-P: transurethral resection of the prostate).

**Table 1 toxins-14-00035-t001:** Improvement in urinary incontinence after detrusor Botox injections in patients with chronic spinal cord injury.

Patient Characteristics	N	Not Improved	Mildly Improved	Moderately Improved	Markedly Improved	*p* Value *
**Male**	154	24 (15.6%)	30 (19.5%)	64 (41.6%)	36 (23.4%)	0.109
**Female**	69	11 (15.9%)	21 (30.4%)	17 (24.6%)	20 (29.0%)	
**Cervical SCI**	89	10 (11.2%)	19 (21.3%)	36 (40.4%)	24 (27.0%)	0.045
**Thoracic SCI**	115	18 (15.7%)	27 (23.5%)	42 (36.5%)	28 (24.3%)	
**Lumbar and sacral**	19	7 (36.8%)	5 (26.3%)	3 (15.8%)	4 (21.1%)	
**Complete SCI**	185	25 (13.5%)	43 (23.2%)	70 (37.8%)	47 (25.4%)	0.221
**Incomplete SCI**	38	10 (28.6%)	8 (21.1%)	11 (28.9%)	9 (23.7%)	
**With AD**	145	28 (19.3%)	36 (24.8%)	45 (31.0%)	36 (24.8%)	0.02
**No AD**	78	7 (9.0%)	15 (19.2%)	36 (46.2%)	20 (25.6%)	
**Hand Function Normal**						
**Partial**	161	28 (17.4%)	40 (24.8%)	53 (32.9%)	40 (24.8%)	0.013
**Incapable ***	23	3 (13.0%)	8 (34.8%)	6 (26.1%)	6 (26.1%)	
	39	4 (10.3%)	3 (7.7%)	22 (56.4%)	10 (25.6%)	

SCI: spinal cord injury, AD: autonomic dysreflexia, * clean intermittent catheterization by care giver.

**Table 2 toxins-14-00035-t002:** Urodynamic parameters of patients with different urinary incontinence improvement following detrusor onabotulinumtoxinA injections.

Urodynamic Parameters	1. Not Improved	2. Mildly Improved	3. Moderately Improved	4. Markedly Improved	*p* Value	
					All	3 + 4
**Age**	43.6 ± 15.1	43.1 ± 14.4	37.6 ± 14.0	38.8 ± 14.8	0.071	0.009
**Duration**	6.9 ± 8.1	7.1 ± 8.0	6.4 ± 7.0	7.3 ±7.9	0.920	0.809
**FSF (mL)**	147 ± 101	120 ± 80.0	114 ± 75.1	107 ± 66.8	0.118	0.072
**FS (mL)**	207 ± 116	163 ± 102	159 ± 102	127 ± 72.6	0.003	0.011
**US (mL)**	228 ± 133	177 ± 113	180 ± 115	134 ± 76.3	0.010	0.018
**Compliance**	59.7 ± 83.8	57.7 ± 74.9	37.4 ± 48.5	34.8 ± 55.0	0.090	0.021
**Pdet (cmH_2_O)**	19.1 ± 16.1	25.8 ±12.6	37.0 ± 17.6	58.5 ± 23.2	All	<0.0001
**Qmax (mL/s)**	7.31 ± 6.87	5.27 ± 7.05	4.59 ± 5.14	4.80 ± 4.72	0.126	0.101
**Volume(mL)**	110 ± 124	68.5 ± 99.1	63.5 ± 82.6	76.2 ± 90.9	0.117	0.242
**PVR(mL)**	203 ± 187	163 ± 168	193 ± 176	126 ± 112	0.065	0.554
**VE (%)**	43.0 ± 39.4	33.8 ± 35.2	30.8 ± 32.1	39.7 ± 35.5	0.269	0.520
**BCI**	55.7 ± 394	52.1 ± 41.9	59.9 ± 32.7	82.5 ± 36.8	All	<0.0001
**BOOI**	4.51 ± 20.0	15.2 ± 14.7	27.8 ± 19.4	48.9 ± 23.0	All	<0.0001

FSF: first sensation of filling, FS: full sensation, US: urge sensation, Pdet: detrusor pressure, Qmax: maximum flow rate, PVR: postvoid residual, VE: voiding efficiency, BC: bladder contractility index, BOOI: bladder outlet obstruction index.

**Table 3 toxins-14-00035-t003:** Baseline videourodynamic characteristics and improvement in urinary incontinence following detrusor onabotulinumtoxinA injections.

VUDS Characteristics	N	Not Improved	Mildly Improved	Moderately Improved	Markedly Improved	*p* Value
**NDO, No DSD**	13	6 (46.2%)	4 (30.8%)	2 (15.4%)	1 (7.7%)	<0.0001
**NDO, with DSD**	150	11 (7.3%)	36 (24.0%)	56 (37.3%)	47 (31.3%)	
**NDO + BND + DSD**	39	6 (15.4%)	9 (23.1%)	18 (46.2%)	6 (15.4%)	
**DU + low compliance**	19	10 (52.6%)	2 (10.5%)	5 (26.3%)	2 (10.5%)	
**DU + ISD**	2	2 (100%)	0	0	0	
**Low pressure**	102	29 (28.4%)	35 (34.3%)	31 (30.4%)	7 (6.9%)	<0.0001
**High pressure**	118	4 (3.4%)	16 (13.6%)	49 (41.5%)	49 (41.5%)	
**Equivocal**	3	2 (66.7%)	0 (0%)	1 (33.3%)	0 (0%)	

NDO: neurogenic detrusor overactivity, DSD: detrusor sphincter dyssynergia, BND: bladder neck dysfunction, DU: detrusor underactivity, ISD: intrinsic sphincter deficiency.

**Table 4 toxins-14-00035-t004:** Conversion of bladder management and improvement in urinary incontinence or voiding condition after bladder management and surgical procedures.

Bladder Management	N *	1. Not Improved	2. Mildly Improved	3. Moderately Improved	4. Markedly Improved	*p* Value *	
						All	3 + 4
**Detrusor Botox/CIC**	108	11 (10.2%)	20 (18.5%)	49 (45.4%)	28 (25.9%)	0.014	0.003
**AE**	29	4 (13.8%)	7 (24.1%)	10 (34.5%)	8 (27.6%)	0.976	0.940
**Ileal conduit**	1	0	0	0	1 (100%)	0.637	0.386
**Urethral Botox**	24	4 (16.7%)	6 (25.0%)	8 (33.3%)	6 (25.0%)	0.987	0.741
**TUI-BN**	9	3 (33.3%)	2 (22.2%)	3 (33.3%)	1 (11.1%)	0.452	0.312
**TUI-P/TUR-P**	6	0	1 (16.7%)	3 (50.0%)	2 (33.3%)	0.840	0.410
**Suburethral sling**	5	2 (40.0%)	1 (20.0%)	2 (40.0%)	0	0.345	0.376
**Cystostomy**	8	3 (37.5%)	2 (25.0%)	1 (12.5%)	2 (25.0%)	0.223	0.266
**Indwelling catheter**	14	1 (7.1%)	4 (28.6%)	6 (42.9%)	3 (21.4%)	0.834	0.821
**CIC alone**	11	3 (27.3%)	2 (18.2%)	1 (9.1%)	5 (45.5%)	0.099	0.753
**External sphincterotomy**	1	0	0	0	1 (100%)	0.637	1.000
**Self-voiding and medication**	88	17 (19.3%)	24 (27.3%)	26 (29.5%)	21 (23.9%)	0.224	

* Patients might have >1 bladder management or surgical procedure. Botox: botulinum toxin A, CIC: clean intermittent catheterization, AE: augmentation enterocystoplasty, TUI-BN: transurethral incision of bladder neck, TUI-P: transurethral incision of the prostate, TUR-P: transurethral resection of the prostate.

**Table 5 toxins-14-00035-t005:** Final overall satisfaction to the bladder management and surgical procedures in chronic spinal cord injured patients.

Bladder Management	N *	Satisfied to Current Status	Acceptable Wish to Change	Not Satisfied but No Change	*p* Value *
**Detrusor Botox/CIC**	108	82 (75.9%)	15 (13.9%)	11 (10.2%)	<0.0001
**AE**	29	24 (82.8%)	2 (6.9%)	3 (10.3%)	0.170
**Ileal conduit**	1	1 (100%)	0	0	1.000
**Urethral Botox**	24	15 (62.5%)	8 (33.3%)	1 (4.2%)	0.060
**TUI-BN**	9	4 (44.4%)	3 (33.3%)	2 (22.2%)	0.355
**TUI-P/TUR-P**	6	6 (100%)	0	0	0.343
**Suburethral sling**	5	3 (60.0%)	2 (40.0%)	0	0.390
**Cystostomy**	8	2 (25.0%)	5 (62.5%)	1 (12.5%)	0.017
**Indwelling catheter**	14	9 (64.3%)	3 (21.4%)	2 (14.3%)	0.889
**CIC alone**	11	7 (63.6%)	2 (18.2%)	2 (18.2%)	1.000
**External sphincterotomy**	1	1 (100%)	0	0	1.000
**Self-voiding and medication**	88	44 (50%)	13 (14.8%)	31 (35.2%)	<0.0001

* Patients might have >1 bladder management or surgical procedure. Botox: botulinum toxin A, CIC: clean intermittent catheterization, AE: augmentation enterocystoplasty, TUI-BN: transurethral incision of bladder neck, TUI-P: transurethral incision of the prostate, TUR-P: transurethral resection of the prostate.

## Data Availability

Data is available on request to the corresponding author.

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
