# Peer review of "Satisfaction with Detrusor OnabotulinumtoxinA Injections and Conversion to Other Bladder Management in Patients with Chronic Spinal Cord Injury"

_toxins, 2022, doi:10.3390/toxins14010035_

Round 1
Reviewer 1 Report
Appropriate patient selection is usually key for successful outcomes of intravesical Botox in NDO. The success rates presented in this manuscript are overall similar to those previously published since approximately 85% of the study population had at least 50% improvement in key clinical parameters, although only 25% of the patients achieved complete cure. However, a look at the treatment options chosen by the patients who discontinued Botox is quite intriguing as at least 40 patients (almost one out of five) opted for surgical procedures aiming at bladder emptying and not urine storage. Additionally, 88 patients (almost 40% of the study population), decided to discontinue Botox injections and resume various voiding techniques. This could actually mean that those patients were possibly treated with the ‘wrong’ treatment when undergoing the Botox injections and, thus, were highly likely to discontinue Botox treatment. Please discuss
What were the clinical and urodynamic characteristics of the patients who discontinued Botox treatment? Were there any differences with those who continued with Botox injections?
How was the severity of reported symptomatic relapse confirmed? Were patients asked to complete a bladder diary or were they submitted to new urodynamic investigation?
At what time-points were the patients followed-up? Please clarify
Author Response
Reviewer #1
Appropriate patient selection is usually key for successful outcomes of intravesical Botox in NDO. The success rates presented in this manuscript are overall similar to those previously published since approximately 85% of the study population had at least 50% improvement in key clinical parameters, although only 25% of the patients achieved complete cure. However, a look at the treatment options chosen by the patients who discontinued Botox is quite intriguing as at least 40 patients (almost one out of five) opted for surgical procedures aiming at bladder emptying and not urine storage. Additionally, 88 patients (almost 40% of the study population), decided to discontinue Botox injections and resume various voiding techniques. This could actually mean that those patients were possibly treated with the ‘wrong’ treatment when undergoing the Botox injections and, thus, were highly likely to discontinue Botox treatment. Please discuss
Reply: Thank you for the comment. We totally agree with your opinion. Appropriate patient selection is the key for successful outcomes of detrusor Botox injection for NDO in chronic SCI patients. In this study 40 (18%) patients opted for surgical procedures aiming at bladder emptying and not urine storage, and 88 (39.5%) finally decided to resume spontaneous voiding, suggesting the treatment strategy in the initial decision to select detrusor Botox was wrong because they could not accept CISC to emptying bladder, therefore, were highly likely to discontinue Botox treatment. This result further implies the importance of patient education and counseling before Botox treatment. (Lines 265-272)
What were the clinical and urodynamic characteristics of the patients who discontinued Botox treatment? Were there any differences with those who continued with Botox injections?
Reply: Thank you for the comments. Our results interestingly showed that patients having NDO without DSD (23.1% with moderate and marked improvement) and those having DU with low compliance (36.8%) or DU with ISD (0%) had less favorable treatment outcomes after detrusor Botox injection. (Lines 125-128). When we analyzed the baseline urodynamic characteristics, more patients who discontinue Botox treatment had NDO without DSD or having DU, compared with those who continued Botox treatment (p = 0.039). These urodynamic results might result in less favorable treatment outcome and influence the decision making in the future bladder management. (Lines 130-135) In fact, most patients who discontinued Botox treatment and returned to self-voiding with incontinence just because the need for CISC had become a burden to their daily life. (Lines 226-228, and 234-236)
How was the severity of reported symptomatic relapse confirmed? Were patients asked to complete a bladder diary or were they submitted to new urodynamic investigation?
Reply: Patients were assessed for decrease in urinary incontinence by their subjective perception of the symptom severity after detrusor Botox injections. (Lines 342-343) Moreover, patients were also assessed according to overall satisfaction with their treatment outcomes in both storage and emptying conditions. (Lines 354-355)
At what time-points were the patients followed-up? Please clarify
Reply: The patients were usually followed up regularly at the SCI clinic every 3 to 6 months after any kind of treatment, (Lines 321-322) and the symptoms were assessed during the proceeding year. (Line 344) The final treatment outcome was assessed by chart review or telephone interview by the end of 2020. (Lines 345-346)
Reviewer 2 Report
Manuscript entitled „ Satisfaction with Detrusor OnabotulinumtoxinA Injections and 2 Conversion to Other Bladder Management in Patients with 3 Chronic Spinal Cord Injury” is an interesting, well-written and well-planned experimental work. However, the text needs some corrections according to the following comments:
Introduction
line 32 – could you explain in the text why NDO frequently occurs in patients with chronic spinal cord injury, what is a pathophysiological mechanism of its presence
line 34 – explain in the text why oral treatment is ineffective in case of urinary incontinence during NDO
line 36 – write in the text all what is known about botulinum toxin, its mechanism of action, how it affects the nervous system, why its use is effective in the treatment of urinary bladder disorders? Please write down everything you need to explain the use of this neurotoxin in the treatment of bladder dysfunction
line 54 – put dot on the end of sentence
Materials and Methods
line 72- what this regularity means -how often patients voiding conditions have been checked.
line 74 – what this baseline means?
line 124 – explain in full name abbreviation SPSS
Figure 1 - Acronyms/Abbreviations/Initialisms should be defined the first time they appear in each of three sections: the abstract; the main text; the first figure or table.
Additionally, the whole section of Materials and Methods should be transferred, and placed after Conclusion section in accordance with the formula specified by the journal.
References
All References must be corrected. All literature items must be prepared in accordance with the journal's rules and policies as follows:
Journal Articles:
- Author 1, A.B.; Author 2, C.D. Title of the article. Abbreviated Journal Name Year, Volume, page range.
Books and Book Chapters:
- Author 1, A.; Author 2, B. Book Title, 3rd ed.; Publisher: Publisher Location, Country, Year; pp. 154–196.
- Author 1, A.; Author 2, B. Title of the chapter. In Book Title, 2nd ed.; Editor 1, A., Editor 2, B., Eds.; Publisher: Publisher Location, Country, Year; Volume 3, pp. 154–196.
Author Response
Reviewer #2
Manuscript entitled „ Satisfaction with Detrusor OnabotulinumtoxinA Injections and Conversion to Other Bladder Management in Patients with Chronic Spinal Cord Injury” is an interesting, well-written and well-planned experimental work. However, the text needs some corrections according to the following comments:
Introduction
line 32 – could you explain in the text why NDO frequently occurs in patients with chronic spinal cord injury, what is a pathophysiological mechanism of its presence
Reply: Thank you for the comment. We have added a statement: due to interruption of spinal tract, lack of central nervous control and regulation, and emergency of C-fiber as the predominant afferent nerves [1]. (Lines 61-63)
line 34 – explain in the text why oral treatment is ineffective in case of urinary incontinence during NDO
Reply: We have added a statement: NDO, is usually difficult to completely eradicated after oral overactive bladder medication because current oral medication for NDO cannot completely block the detrusor contractility [2]. (Lines 63-66) In patients who still bother by urinary incontinence, detrusor injection of onabotulinumtoxinA (Botox) can provide a chance to regain urinary continence [3]. (Lines 66-68)
line 36 – write in the text all what is known about botulinum toxin, its mechanism of action, how it affects the nervous system, why its use is effective in the treatment of urinary bladder disorders? Please write down everything you need to explain the use of this neurotoxin in the treatment of bladder dysfunction
Reply: Thank you for the comment. We have added a paragraph to describe the mechanism of the use of Botox in the treatment of NDO in chronic SCI patients. (Lines 70-78)
line 54 – put dot on the end of sentence
Reply: Thank you for the comment. We have revised it.
Materials and Methods
line 72- what this regularity means -how often patients voiding conditions have been checked.
Reply: Thank you for the comment. Patients were regularly followed up every 3-6 months. (Lines 321-322)
line 74 – what this baseline means?
Reply: Thank you for the comment. When the episodes and severity of urinary incontinence returned to baseline condition. (Lines 323-324)
line 124 – explain in full name abbreviation SPSS
Reply: We have added the full name of SPSS, Statistical Product and Service Solution. Line 381)
Figure 1 - Acronyms/Abbreviations/Initialisms should be defined the first time they appear in each of three sections: the abstract; the main text; the first figure or table.
Reply: Thank you for the comment. The abbreviations of the figure 1 have been added in the figure legend. (Lines 523-525)
Additionally, the whole section of Materials and Methods should be transferred, and placed after Conclusion section in accordance with the formula specified by the journal.
Reply: Thank you for the comment. The whole section of Materials and Methods has been transferred, and placed after Conclusion section.
References
All References must be corrected. All literature items must be prepared in accordance with the journal's rules and policies as follows:
Reply: Thank you. The references writing has been corrected in accordance with the journal's rules and policies.
Reviewer 3 Report
This is a rather confirmative study with a retrospective study design and a quiet large cohort making it interesting. From a methodological point of view there are some major drawbacks, mainly lack of information to make it possible to the reader to judge the value of the observation:
- Where come the data from, from a registry or from an electronic patient file or a written patient file?
- Assessment of continence: 4 grades, these are the cornerstone information in the result section and are not a validated tool. Was this assessed during the consultation (through standardized questioning?) or posthoc judged through the authors from what is written in the patient file?
- There is no information at all on missing values
The main finding according to the conclusion is the 48.4% continuation rate, in the introduction the authors refer to ref 14 : “Approximately 60% of the patients treated with intradetrusor onabotulinumtoxinA injections for refractory neurogenic detrusor overactivity continue this therapy long term with good therapeutic effects”. However this is a much smaller cohort with mixed neurogenic diseaeses including MS, Spina bifida,… Another medium longterm followup study (your citation 12, not cited in the intro) showed The 10-year discontinuation-free and failure-free survival rates were 49.1% and 73%, respectively. (Baron M, J Urol 2019) which is very comparable to this study. This makes the main conclusion as a confirmative message.
Author Response
Reviewer #3
This is a rather confirmative study with a retrospective study design and a quiet large cohort making it interesting. From a methodological point of view there are some major drawbacks, mainly lack of information to make it possible to the reader to judge the value of the observation:
Where come the data from, from a registry or from an electronic patient file or a written patient file?
Reply: Thank you for the comment. The patients had been treated and followed up in the Urology department of the hospital from 2000 to 2020. (Lines 303-305) The final treatment outcome was assessed by chart review or telephone interview by the end of 2020. (Lines 345-346)
Assessment of continence: 4 grades, these are the cornerstone information in the result section and are not a validated tool. Was this assessed during the consultation (through standardized questioning?) or posthoc judged through the authors from what is written in the patient file?
Reply: All SCI patients had urinary incontinence in this study. The severity of urinary incontinence is usually difficult to evaluate because most of them do not have bladder sensation. The assessment of urinary incontinence improvement was graded to four grades. (Line 346) The condition of incontinence improvement was described in the Methods, according to the reduction of urine leakage quantity and diaper use. (Lines 346-352) There was no validation for this grade, but it is rational.
There is no information at all on missing values
Reply: Thank you for the comment. Patients who were lost to follow-up and the data unavailable to assess were not included in this study. (Lines 368-369)
The main finding according to the conclusion is the 48.4% continuation rate, in the introduction the authors refer to ref 14 : “Approximately 60% of the patients treated with intradetrusor onabotulinumtoxinA injections for refractory neurogenic detrusor overactivity continue this therapy long term with good therapeutic effects”. However this is a much smaller cohort with mixed neurogenic diseaeses including MS, Spina bifida, Another medium longterm followup study (your citation 12, not cited in the intro) showed The 10-year discontinuation-free and failure-free survival rates were 49.1% and 73%, respectively. (Baron M, J Urol 2019) which is very comparable to this study. This makes the main conclusion as a confirmative message.
Reply: Thank you for the comment. We agree that the conclusion of this study provides a confirmative message. We also stated in the discussion section. A 10-year follow-up of a large cohort of patients with SCI receiving detrusor Botox injections found that approximately 50% were still receiving injections for their NDO [24]. The results of this study confirmed that the persistence rate of detrusor Botox treatment for NDO in chronic SCI patients was consistent in different studies with long-term follow-up [19,20]. (Lines 184-188)
Round 2
Reviewer 1 Report
Thank you for addressing this reviewer’s comments. It is revealing the authors acknowledge that a significant 40% of the study cohort actually chose Botox for the wrong reasons. And eventually, 18% of the study patients opted for surgical procedures aiming at more efficient bladder emptying, while several resumed ‘spontaneous’ bladder emptying techniques, clearly suggesting that the bladder emptying phase was more important for their quality of life than the storage phase. This became more obvious as their satisfaction following surgical emptying procedures was higher than satisfaction for the treatment of incontinence. Therefore, I think we could assume that the study cohort suffered from heavy selection bias, and I wonder whether a separate analysis of results should have been (or should be) done for the 40% of patients who were, after all, unhappy with CISC, although acceptance of the possibility of the post-Botox need for CISC is a ‘must’ in the counselling of every patient who will have bladder Botox injections. In addition, it should be made clear in Conclusions that results have been biased by the inappropriate patient selection.
Author Response
Reviewer #1
Thank you for addressing this reviewer’s comments. It is revealing the authors acknowledge that a significant 40% of the study cohort actually chose Botox for the wrong reasons. And eventually, 18% of the study patients opted for surgical procedures aiming at more efficient bladder emptying, while several resumed ‘spontaneous’ bladder emptying techniques, clearly suggesting that the bladder emptying phase was more important for their quality of life than the storage phase. This became more obvious as their satisfaction following surgical emptying procedures was higher than satisfaction for the treatment of incontinence. Therefore, I think we could assume that the study cohort suffered from heavy selection bias, and I wonder whether a separate analysis of results should have been (or should be) done for the 40% of patients who were, after all, unhappy with CISC, although acceptance of the possibility of the post-Botox need for CISC is a ‘must’ in the counselling of every patient who will have bladder Botox injections. In addition, it should be made clear in Conclusions that results have been biased by the inappropriate patient selection.
Reply: Thank you for the comment. We have added statements regarding the patient’s predominant urological symptoms and bladder management selection. (Lines 249 to 255)
We also add statements regarding the inappropriate patient selection resulting in unsatisfactory treatment outcome and discontinue detrusor Botox injection in the discussion section (Lines 272 to 281)
We have analyzed the baseline demographics and urodynamic results in these 88 patients who discontinued detrusor Botox injection and resumed self-voiding with medication. The male/female ratio (67/21) was higher than the rest of patients (87/48), cervical SCI predominant (39/49 vs 50/135), more patients with DU (13/88 vs 8/135), however, the urodynamic parameters did not significantly different from the rest of SCI patients.
The study is a retrospective analysis of the treatment outcome and discussing the factors influencing bladder management in patients with chronic SCI. Inappropriate patient selection has been addressed in the discussion section. Appropriate patient selection is the key for successful outcomes of detrusor Botox injection for NDO in chronic SCI patients. (Lines 270-271).
We also add statement that results have been biased by the inappropriate patient selection, (Lines 303-304) and the causes of discontinuing Botox injection include inappropriate patient selection for detrusor Botox injection. (Lines 305-306)
Reviewer 3 Report
no comment
Author Response
Thank you